



Soil carbon fractions and enzyme activities under different vegetation types on
the Loess Plateau of China
Haixing Zhang [a], Quanchao Zeng [a], Shaoshan An [a, b], Yanghong Dong [a], Frédéric
Darboux [c]
a College of Natural Resources and Environment, Northwest A&F University, 712100, P.R. China
b State key laboratory of soil erosion and dryland farming on the Loess Plateau, Institute of Soil
and Water Conservation, Northwest A&F University, 712100, P.R. China
c INRA, Laboratoire Sols et Environnement, UMR 1120, Vandœuvre-lès-Nancy, F-54518, France
Corresponding author: Tel.: +86 29 87012871; Fax: +86 29 87012210.
*E-mail address:*shan@ms.iswc.ac.cn (S.S. An).



**Abstract:**
Vegetation restoration was effective way of protecting soil erosion and water
conservation on the Loess Plateau. Carbon fractions and enzyme activities were
sensitive parameters for assessment of soil remediation through revegetation. Forest,
forest steppe and grassland soils were collected at 0-5 cm and 5-20 cm soil layers in
Yanhe watershed, Shaanxi Province. Urease, sucrase, alkaline phosphatase, soil
organic carbon (SOC), microbial biomass carbon (MBC), easily oxidized organic
carbon (EOC) and dissolved organic carbon (DOC) were measured. The results
showed that carbon fraction contents and enzyme activities in the same soil layer
followed the order that forest was higher than others. Carbon fraction contents and
enzyme activities appeared that the 0-5 cm was higher than 5-20 cm soil layer. In
addition, correlation analysis showed that urease activity was related to SOC, MBC,
EOC and DOC at 0-5 cm layer; it was correlated with SOC, MBC and EOC at 5-20
cm layer. Sucrase activity had significant positive relationship with SOC, MBC and
EOC. Alkaline phosphatase activity was related to EOC and DOC at 0-5 cm layer; it
was correlated with MBC and EOC at 5-20 cm layer. The CCA reflected the
relationship between sucrase activity and SOC. The contributions from the various
forms of carbon fractions and enzyme activities as evaluated by the canonical
coefficient of CV were on the order of SOC > DOC > MBC > EOC; sucrase > urease >
alkaline phosphatase. Vegetation type was an important factor influencing the
variation of soil enzyme activities and carbon fractions on the Loess Plateau.
**Key Words:** vegetation types; soil organic carbon; soil microbial biomass carbon; soil





easily oxidized organic carbon; soil dissolved organic carbon; soil enzyme activities

## 1. Introduction

Land degradation and soil erosion are serious problems in the Loess Plateau of China
(Fu et al., 2005; Zheng et al., 2005). Zheng et al., (2005) reported that the nutrient loss
was strongly related to erosion patterns and erosion intensity. Since 1999, the Grain
for Green Project had been implemented in the Loess Plateau. It induces improvement
in vegetation conditions may benefit soil erosion alleviation and carbon sequestration
in the Loess Plateau (Wang et al., 2011; Zhou et al., 2012). Studies of revegetation
after farmland abandonment in the Loess Plateau of China indicated that soil physical
properties are closely related to the vegetation recovery stages (Li and Shao, 2006;
Zuo et al., 2009). Some researchers stress that the vegetation restoration in Loess
Plateau is very important for soils health, a long-term experiment show that
integrative measures restore forests and stop soil erosion on the severely eroded bare
land (Zhang et al., 2004); Chen and Cai (2006) found that reduction of reclamation
rate and the increase of tree and grass vegetation could control soil erosion in the
sandy and coarse sandy areas; and when human activities destroyed secondary forests,
soil erosion increased (Zheng, 2006). Recently, some studies have concentrated on the
vegetation restoration, for instance, Jiao et al., (2011) found that revegetation had
positive effects on the soil physical properties. In the protected vegetation areas,
relative humidity of air increased and wind velocity is greatly reduced. Additionally,
bulk density of the surface layer (0-20 cm) significantly decrease while soil porosity,
water-holding capacity, aggregate stability, and saturated hydraulic conductivity





significantly increase. SOC stocks are increased by 19% in the surface soil layer at
0-20 cm soil depth from 1998 to 2006, because of the vegetation restoration in the
Loess Plateau (Wang et al., 2011).
The soil carbon fractions include soil organic carbon (SOC), microbial biomass
carbon (MBC), easily oxidized organic carbon (EOC) and dissolved organic carbon
(DOC). Soil organic carbon and enzyme activities are indicators of soil fertility
(Gregorich et al., 1994; Lagomarsino et al., 2011). SOC storage is estimated about
two and three times the size of carbon pools in the atmosphere and vegetation,
respectively (Jobbágy and Jackson, 2000; Lal, 2004). SOC stocks in 0-30 cm soil
layer are highly variable among the vegetation communities (Yimer et al., 2006). SOC
plays a key role in the global C cycle (Noble et al., 2000) and as indicator of soil
quality (Gregorich et al., 1994); it is also an important component of agricultural soils
(Fang et al., 2012). Labile organic carbon (MBC, EOC and DOC) plays an important
character in short-term turnover of soil nutrients and provides energy for microbes
(Piao et al., 2000); it has a higher activity for microbes (Shen et al., 1999). Soil MBC
is used as an indicator of changes in soil organic matter (Jenkinson, 1988; Saffigna et
al., 1989), it generally represents 2–3% of soil organic C (Anderson and Domsch,
1989). EOC is an indicator of soil labile organic carbon (Biederbeck et al., 1994).
DOC is sensitive to soil quality and fertility transformations, hence it can better reflect
the soil physical and chemical properties (Lu et al., 2006).
Enzyme activities can express soil quality by providing useful linkages between
the microbial community structures and the environmental factors (Zhang et al., 2015).



Ecoenzymatic stoichiometry, microbial respiration, and organic matter decomposition
are responsive to resource availability and the environmental drivers of microbial
metabolism (Hill et al., 2014). Large numbers of these enzymes are expressed and
released into the environment by microorganisms in response to environmental
signals (Sinsabaugh et al., 2009). The soil enzyme activities can be crucial in
detecting differences among forest, monoculture and intercropping (de Medeiros et al.,
2015). Microbial enzyme allocation is sensitive to differences in nutrient limitation
(Moorhead et al., 2012). However, there is a lack of information on the relationship
between soil carbon and enzyme activities for soils with different vegetation types.
We advanced the following three hypotheses.
H1: both carbon fraction contents and enzyme activities in the same soil layer are
higher for forest than for forest steppe and grassland.
H2: carbon fraction contents and enzyme activities under all vegetation soils are
higher in the surface layer than in the underlying layer.
H3: different carbon fractions have different effects on enzyme activities in soil.
To this end we investigated four carbon fractions and three enzyme activities under
various type vegetations considered in our experiments.
**2. Materials and methods**
2.1 Study sites
The field site (107°41′～110°31′E, 35°21′～37°31′N) is located in Yanhe watershed,
northern Shaanxi Province, China (Table 1). It belongs to the hilly-gully part of the
Loess Plateau and has a total area of 37029 km$^2$. Its average elevation is about 1000 m.



It has a continental arid to semi-arid climate, with an annual average frost-free period
of 170 d, an annual average temperature of 9.2 ℃, an annual average sunshine
duration of 2500 h, and an annual average precipitation of about 500 mm (CCSLC,

2000).

2.2 Soil collection and processing
Soils were collected in August, 2013, on three typical vegetation types (grassland,
forest steppe and forest). For each vegetation type, four representative plant
communities were chosen (Table 1), and, as replicates, three sampling areas were
defined in the field for each representative plant community. In each representative
plant community, three sampling plots were delineated. The sizes of the sampling
plots were: 20×20 m for forest, 5×5 m for forest steppe and 1×1 m for grassland.
Within each plot, based on an S-shaped sampling pattern, the incompletely-degraded
litter was removed and 9 sub-samples were simultaneously and randomly collected
then mixed them in the same bag which as a representative soil sample, separately at
0-5 cm and 5-20 cm depth. The representative soil sample was split into two parts,
one was stored intact at -20 ℃ in order to determine carbon fractions and enzyme, and
the other was air-dried for measuring soils' physics and chemical properties.
2.3 Methods
2.3.1 Carbon assay
SOC was determined by wet digestion with a mixture of potassium dichromate and
concentrated sulfuric acid (ISSSC, 1981). The soil organic matter is various in
different type soil, 0.1 g of air-dried soil was weighed in boiling tube, then 5 ml





$K_2CrO_4$ and 5 ml concentrated sulfuric acid were added and shaken well. The sample
was put into the 185-190 ℃ paraffin oil bath. Soil sample was taken out after boiling
for 5 min. After cooling, the substance in the tube was transferred into an Erlenmeyer
flask, and 2-3 drops of phenanthroline indicator were added before being titrated with
$FeSO_4$ solution. The color of the solution changed from orange to blue and in the end
it turned brick red with $FeSO_4$ titration solution adding.

The MBC was measured using the chloroform fumigation–extraction method

(Ross, 1990). The soil sample was taken from -20 ℃ freezer and thawed, 100 g soil
which adjusted to 60% of field capacity was added to a 500 ml jar, incubated for 7
days at 25 ℃. The soil sample was exposed to chloroform vapor in a vacuum
desiccator at 25 ℃ for 24 h .After chloroform fumigation, the total carbon content was
determined in the 0.5 M $K_2SO_4$ extract. Determination of carbon content used the
TOC-1020A organic carbon analyzer (Phoenix 8000, USA).

Soil EOC was measured using a slightly modified version of the light group

organic compound separation method of (Janzen et al., 1992). A sample containing 15
mg of carbon was put into a 100 ml centrifuge tube. 25 ml of 333 mMol/L potassium
permanganate was added and shaken for an hour, and then centrifuged at 4000 rpm
for 5 min. The supernatant was diluted with deionized water at 1:250, and then the
absorbances of the blank and soil sample were determined by spectrophotometry at
565 nm (TOC-1020A organic carbon analyzer, Phoenix 8000, USA). By comparing
the absorbances of the blank and soil sample, the change of the potassium
permanganate concentration was calculated, and then the amount of oxidated carbon.



Soil DOC was measured by $K_2SO_4$ leaching-TOC method (Murphy et al., 2000).
10 g of air-dried 2-mm-sieved sample was weighed, and distilled water was added at a
ratio of 2:1. After shaking for 30 min at 25 ℃ constant temperature, filtering was
carried out on a membrane filter and was determined using the TOC-1020A organic
carbon analyzer (Phoenix 8000, USA)
2.3.2 Enzyme assay
Urease activity was determined by indophenol blue colorimetry (Guan et al., 1991). 5
g of air-dried 2-mm-sieved soil was added into 50 ml Erlenmeyer and 1 ml toluene
was added. Then it was left for 15 min before adding 10 ml of 10% urea solution and
20 ml of sodium citrate butter, and shook well. The sample was subsequently
incubated at 37 ℃ for 24 h and then diluted to 50 ml with 37 ℃ distilled water. The
suspension was filtered and 1 ml of the extract was added to a 50 ml flask with 4 ml
sodium phenol solution and 3 ml sodium hypochlorite solution. After shaking well
and then let it rest for 20 minutes, the released ammonium was extracted with
potassium chlorite solution. The ammonium was quantified colorimetrically with a
spectrophotometer (2800 UV/VIS) at 578 nm.
Sucrase activity was determined by 3, 5 - dinitrosalicylic acid colorimetry (Guan
et al., 1991). 5 g of soil was weighed in an Erlenmeyer and 5 drops of toluene was
added before being gently shaken. Let it rest for 10 minutes, then 15 mL of 8%
glucose solution and 5 mL of phosphate buffer at pH 5.5 were added. The sample was
subsequently incubated at 37 ℃ for 24 h. The suspension was filtered and 1 ml of the
extract was added to a 50 ml flask. 3 mL of 3, 5 - dinitrosalicylic acid was added



before a 5 min heating in a boiling water bath. It was then diluted to 50 ml. The
sample colorimetric measurement was determined in spectrophotometer (2800
UV/VIS) at 508 nm.

Alkaline phosphatase was determined using disodium phenyl phosphate method

(Guan et al., 1991). 2 g sample was weighed in 50 ml tube, then 2.5 ml of toluene and
20 ml of 0.5% disodium phenyl phosphate were added. The sample was subsequently
incubated at 37 ℃ for 24 h. 100 ml of 0.3% aluminum sulfate solution was added
before filtering. 3 ml of filtrate was added into a 50 ml flask. The sample colorimetric
measurement was determined in spectrophotometer (2800 UV/VIS) at 660 nm.
2.4 Data analysis
Data were processed by Excel 2010; statistical analyses were carried out with SPSS
19.0 and plotting by Origin Pro. 8.0. One-way ANOVA conducted by the Scheffe test
($p<0.05$) was used to compare the differences among vegetation types.

A canonical correlation coefficients analysis (CCA) was carried to assess the

relationship between two datasets: soil carbon fractions and enzyme activities. The
CCA is designed to identify linear combinations of variables in one dataset that
account for the greatest variation in a linear combination of variables for the other
dataset (Lattin et al., 2003). In this study, the CCA was performed using the soil
carbon fractions and enzyme activities, three pairs of canonical variates (CVs) were
extracted. The $U_1$ and $V_1$ refer to the first group equation between soil carbon
fractions canonical variate (X-CV) and the enzyme activities canonical variate (Y-CV).
The indices of the X-CV were: SOC (X1), MBC (X2), EOC (X3), and DOC (X4).



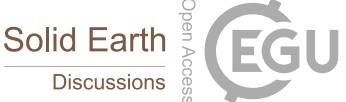

The indices of the Y-CV were: urease activity (Y1), sucrase activity (Y2), alkaline
phosphatase activity (Y3).
**3. Results**
3.1 Physical and chemical properties depending on the vegetation type
Vegetation types had great effects on the soil basic physical and chemical properties.
The bulk density was significantly different between forest and grassland (Table 2).
Forest steppe vegetation's bulk density was significant difference between the 0-5 cm
and 5-20 cm soil layers. The pH was no significant difference between the two layers
of a given vegetation type and it was significantly different between forest and
grassland in both soil layers ($p<0.05$). The total N concentration of forest was
significantly higher than the total N of both forest steppe and grassland, in both soil
layers. The total P concentration of grassland vegetation was significantly lower than
for both forest and forest steppe, in both soil layers. There was no significant
difference in the same vegetation's total P and total N concentration between the two
soil layers. No significant difference in soil organic matter was seen between forest
steppe and grassland, while soil organic matter of forest was significantly higher.
Forest steppe and grassland vegetation's soil organic matter was significantly different
between the 0-5 cm and 5-20 cm soil layers.
3.2 Soil carbon fractions depending on the vegetation type
The SOC, MBC and EOC contents of forest soils were significantly different from
both forest steppe and grassland soils, and there was no significant difference between
forest steppe and grassland vegetation (Fig. 1). The DOC concentration of forest



vegetation was significant difference to grassland vegetation at 0-5 cm soil layer, and
grassland vegetation also was significant difference to both forest steppe and forest
vegetation at 5-20 cm soil layer. Forest steppe and grassland vegetation's SOC and
EOC concentrations were significant difference between the upper and lower soil
layers, except for forest vegetation. There was no significant difference among forest,
forest steppe and grassland vegetation's MBC concentration between the two layers.
Forest and forest steppe vegetation's DOC concentrations were significantly different
between the upper and lower soil layer, except for grassland vegetation. SOC contents
at 0-5 cm soil layer was more 6.73, 3.13 and 1.29 $g\cdot kg^{-1}$ than 5-20 cm soil layer under
forest, forest steppe and grassland; MBC contents at upper soil layer was more 227.44,
102.94 and 62.05 $mg\cdot kg^{-1}$ than lower soil layer under forest, forest steppe and
grassland; EOC contents at upper soil layer was higher 2.24, 0.31 and 0.21 $g\cdot kg^{-1}$ than
lower soil layer under forest, forest steppe and grassland; DOC contents at upper soil
layer was higher 166.06, 122.07 and 44.97 $mg\cdot kg^{-1}$ than lower soil layer under forest,
forest steppe and grassland respectively.
3.3 Soil enzyme activities depending on the vegetation type
For the 0-5 cm soil layer, the urease activity of forest vegetation was significantly
different from forest steppe and grassland vegetations (Fig. 2A), while the sucrase
activity and the alkaline phosphatase activity of forest vegetation were significantly
different from forest steppe and grassland vegetations (Fig. 2B & 2C). For the 5-20
cm soil layer, the urease was significantly different between forest and grassland
vegetations (Fig. 2A), while the sucrase activity was non-significantly different



among forest, forest steppe and grassland vegetations (Fig. 2B). The alkaline
phosphatase activity was significant difference between forest and grassland
vegetation (Fig. 2C). Forest and forest steppe vegetation's urease and alkaline
phosphatase activities were significantly different between the upper and lower soil
layers, except for grassland vegetation (Fig. 2A & 2C). However, forest, forest steppe
and grassland vegetation's sucrase activities were not significantly different between
the upper and lower soil layers (Fig. 2B). The urease activity of all vegetation type
soils appeared that the upper more than lower soil layer, and forest, forest steppe and
grassland increased 35.94, 58.44 and 46.55 percentage which increased 0.46, 0.45 and
0.27 $mg^{.}kg^{-1}$ respectively. The sucrase activity of all vegetation type soils appeared
that the upper more than lower soil layer, and forest, forest steppe and grassland
increased 53.74, 31.66 and 29.19 percentage which increased 8.9, 3.59 and 2.84
$mg^{.}kg^{-1}$ respectively. At 0-5 cm soil layer, sucrase activity of forest vegetation was
1.71 and 2.03 times compared with forest steppe and grassland vegetation,
analogously at 5-20 cm soil layer, sucrase activity of forest vegetation was 1.46 and
1.70 times by comparing with forest steppe and grassland vegetation. Soil alkaline
phosphatase activity of all vegetation type soils appeared that the upper more than
lower soil layer, and forest, forest steppe and grassland increased 27.58, 33.84 and
28.26 percentage which increased 0.91, 0.89 and 0.39 $mg^{.}kg^{-1}$ respectively. At 0-5 cm
soil layer, alkaline phosphatase activity of forest vegetation was 1.20 and 2.38 times
compared with forest steppe and grassland vegetation, analogously at 5-20 cm soil
layer, alkaline phosphatase activity of forest vegetation was 1.25 and 2.39 times by





comparing with forest steppe and grassland vegetation.
3.4 Correlations between soil carbon fraction contents and enzyme activities
3.4.1 Correlations between soil carbon fraction contents and enzyme activities of
different vegetation types at 0-5 cm soil layer
At 0-5 cm soil layer under all vegetations, soil urease activity was significant
correlated extremely with SOC, MBC, EOC and DOC which correlation coefficient
were 0.823, 0.787, 0.775 and 0.886. Soil sucrase activity was positively significant
correlated with SOC, MBC and EOC which correlation coefficient was 0.907, 0.877
and 0.818, there was non-significant correlation with DOC. Soil alkaline phosphatase
activity was positively significant correlated with DOC which correlation coefficient
was 0.727, and it also was significant correlated with EOC which correlation
coefficient was 0.588, and there was non-significant correlation with SOC and MBC
(Table 3).
3.4.2 Correlations between soil carbon fraction content and enzyme activity of
different vegetation types at 5-20 cm soil layer
Soil urease activity was positively significant correlated with SOC which correlation
coefficient was 0.762, meanwhile, it was significant related to MBC and EOC which
correlation coefficient was 0.633 and 0.621, however, there was non-significant
correlation with DOC. Soil sucrase activity was positively significant correlated with
SOC and MBC which correlation coefficient was 0.759 and 0.840, simultaneously, it
was also significant related to EOC which correlation coefficient was 0.593, there was
non-significant correlation with DOC. Soil alkaline phosphatase activity was

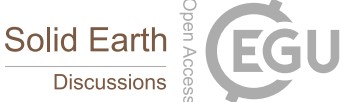

significant related to MBC and EOC which correlation coefficient was 0.656 and
0.600 (Table 4).

The CCA was performed using the soil carbon fractions and enzyme activities,

and three pairs of canonical variates (CVs) were extracted. The canonical correlation
between the first soil carbon fractions canonical variate (X-CV$_1$) and the first enzyme
activities canonical variate (Y-CV$_1$) was significant (R=0.964; $P$<0.001). This first
canonical variate mainly reflected the relationship between the sucrase activity and
SOC. Around 70% of the variance in the Y-CV$_1$ was explained by the X-CV$_1$ (Table 3).
The contributions from the various forms of carbon fractions as evaluated by the
canonical coefficient of CV were on the order of SOC > DOC > MBC > EOC. The
enzyme activities were on the order of sucrase > urease > alkaline phosphatase.
**4. Discussion**
4.1 Soil carbon fraction of different type vegetations
Various factors influence on SOC such as the climate, soil, vegetation, and human
disturbance (Solomon et al., 2007). There are differences among soil carbon fractions
in various type vegetations, due to the diverse restoration years, stages and types of
vegetations (Novara et al., 2015). Meanwhile, decomposition also the main reason, it
is a fundamental ecosystem process and a key ecological process that controlled
nutrient availabilities to plants in terrestrial ecosystems (Moorhead et al., 1996).
Johnsen et al. (2013) found that the amount of C entering the soil through greater
forest litter and belowground biomass production. Decomposition in response to
variations in litter quality and key parameter values (Moorhead et al., 2012), and the





first step in detritus decomposition results from the activity of enzymes produced by
soil microbes (Wallenstein et al., 2009). The soil physical and chemical properties
regulate decomposition rates (Xu et al., 2016). The soils of the study are sampled
from different vegetation types, therefore, their decomposition rates are various and
the carbon pools also are different (Xu et al., 2016). Soil organic matter represents the
largest terrestrial pool for carbon storage (ParrasAlcántara et al., 2015). About
three-fourth of organic carbon contained in terrestrial ecosystems are found in soil
organic matter and plant litters (Schlesinger, 1997; Lal, 2008). Organic matter
decomposition is responsive to resource availability and the environmental drivers of
microbial metabolism (Hill et al., 2014). The decomposition of plant litter may be the
biosphere's most complex ecological process (Sakamoto and Oba, 1994).
Microorganism plays an important role in forest soil carbon and nutrient
transportation (Lipson et al., 2002). Microbial communities are mainly influenced by
local environmental properties (Fierer and Jackson, 2006). Any changes in the
microbial biomass may affect the cycling of carbon (Saffigna et al., 1989; Clein and
Schimel, 1995; Stone et al., 2014), for example, Holt (1997) found that MBC was
lower in the soils of the area that had been subjected to poor grazing management, it
was significantly higher in vegetated soils than in the unvegetated control (Sanaullah
et al., 2011). Soil depth has a highly significant effect on the microbial communities
(Li et al., 2014; Stone et al., 2014). Soil C, MBC are highly correlated with each other
across all soil and forest types and depth increments (Stone et al., 2014), it is
consistent with our conclusion. Result is that different carbon fraction contents at



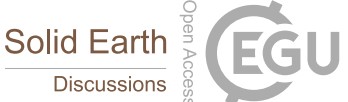

upper soil layer are higher than at the lower.
4.2 Soil enzyme activity of different type vegetations
The study results show that enzyme activity at 0-5 cm soil layer is higher than at 5-20
cm soil layer, mainly due to the large stocks of litter leaves, plants and animal residues
quantity or species at upper layer soil. The soil quality would also affect enzyme
activities (Bandick and Dick, 1999; Zhang et al. 2015), adequate nutrients through
degrading detritus and well soil air permeability make soil microorganism thriving
and enzyme activity higher (Bastida et al., 2013). With the soil layer deepen, soil air
permeability go to worse nutrients made microbial metabolism slowly, these wrong
soil conditions would affect enzyme activities. The soil microbial activity at upper soil
layer is stronger than at the lower, with the microbial activity increasing, the enzyme
activity become higher. Surface soil has adequate excreta from plants, animals and
microorganisms, the physiological activity of upper layer soil is stronger and make
soil released more enzymes. Thereby, enzyme activity declines exponentially with
depth (Stone et al., 2014). Different land used treatments has an influence on soil
enzyme activity (de Medeiros et al., 2015), and enzyme activity associate with plant
litters (Sinsabaugh, 2010). Forest had more plant litters and soil microorganism than
the others, soil enzyme activity is higher under forest vegetation. The paper assume
that various carbon fractions have different effects on soil enzyme activities.
4.3 Relationship between soil carbon fraction and enzyme activity
Majority kinds of soil carbon fractions are sources of microorganism, and have
different effectiveness. Stone et al. (2014) observed strong and interrelated gradients



in soil C, microbial biomass and enzyme activities with depth. Enzymes activities per
unit of total organic carbon and MBC are more important in explaining differences
between soils than absolute enzyme activities in sandy entisol (de Medeiros et al.,
2015). Studies of MBC and enzyme activities provide information on the biochemical
processes occurring in the soil and soil biological parameters may have a potential as
early and sensitive indicators of soil ecological stress and restoration (Dick et al.,
1992; Demisie et al. 2014). Activities of the enzymes are calculated by dividing
enzyme activities by the MBC (Waldrop et al., 2000; Waring et al., 2014). Enzymatic
activity is related to carbon dynamics in soil and can indicate the incorporation of
labile carbon in soil (Kang et al., 1998; Sardans et al., 2008; Lagomarsino et al., 2011).
Enzymes participated in the transformation process of soil nutrients, in the soil
environment, enzyme activity plays vital role on soil microbial activity and soil
quality (Dick, 1994; Masto et al., 2006). Shao et al. (2015) found that there was no
significantly correlated between SOC and urease, while MBC and DOC were
significantly positively correlated with urease. Therefore, Enzyme activity and carbon
fraction has influence each other on conversion and circulation (Plaza et al., 2004;
Mandal et al., 2007).
**5. Conclusions**
Through analysis and research relationship between soil carbon fractions and enzyme
activities of different type vegetation soils in northern Shaanxi Province Loess Plateau,
it revealed the influence of different carbon fractions on soil enzyme activities, the
conclusions were as followings. First, the concentrations of SOC, MBC, EOC and





DOC under different type vegetation soils showed that forest was higher than forest
steppe and grassland at the same soil layer. And these carbon fractions concentration
of all vegetation type soils appeared that the upper was higher than the lower. Second,
the patterns of the enzyme activities were similar to soil carbon fractions. Third,
correlation analysis showed that the SOC, MBC, EOC and DOC influenced on the
urease activities; SOC, MBC and EOC affected sucrase activities; MBC, EOC and
DOC influenced on alkaline phosphatase activities. The CCA reflected the
relationship between the sucrase activity and SOC. The contributions from the various
forms of carbon fractions as evaluated by the canonical coefficient of CV were on the
order of SOC > DOC > MBC > EOC. The enzyme activities were on the order of
sucrase > urease > alkaline phosphatase. In conclusion, vegetation type was an
important factor influence the variation of soil enzyme activities and carbon fractions
on the Loess Plateau.
**Acknowledgments**
This study was supported by the National Natural Sciences Foundation of China

(41671280, 41171226).

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

Table 1



The basic information of different vegetation types sampling points

| Site | Vegetation type | Dominant vegetation |
|---|---|---|
|  |  | *Platycladus orientalis(L.) Franco* |
| Fuxian | Forest | *Quercus liaotungensis* |
|  |  | *Pinus tabulaefrmis* |
|  |  | *Robinia pseudoacacia L.* |
|  |  | *Sophora viciifolia* |
| Ansai | Forest steppe | *Robinia pseudoacacia L.* |
|  |  | *Compositae* |
|  |  | *Hippophae rhamnoides Linn* |
|  |  | *Compositae* |
| Lian Daowan | Grassland | *Artemisia giraldii Pamp* |
|  |  | *Xeric phragmitesaustralis* |
|  |  | *Thymus mongolicus Ronn & Compositae* |






Table 2
Basic physical and chemical properties of the soils for the four vegetation types

| Vegetation type | soil layers | Bulk density (g.cm-3) | Soil pH | Total N (g·kg⁻¹) | Total P (g·kg⁻¹) | Soil organic matter (g·kg⁻¹) |
|---|---|---|---|---|---|---|
| Forest | 0-5 cm | 0.95 ±0.17Ba | 7.93 ±0.12Ba | 1.81 ±0.54Aa | 0.55 ±0.04Aa | 34.82 ±12.94Aa |
| | 5-20 cm | 1.11 ±0.07Ba | 8.02 ±0.12Ba | 1.20 ±0.24Aa | 0.54 ±0.02Aa | 23.23 ±5.68Aa |
| Forest steppe | 0-5 cm | 1.11 ±0.05ABb | 8.11 ±0.08ABa | 0.89 ±0.32Ba | 0.55 ±0.02Aa | 11.88 ±2.00Ba |
| | 5-20 cm | 1.21 ±0.03ABa | 8.21 ±0.06ABa | 0.49 ±0.09Ba | 0.53 ±0.03Aa | 6.48 ±0.46Bb |
| Grassland | 0-5 cm | 1.26 ±0.03Aa | 8.25 ±0.04Aa | 0.57 ±0.08Ba | 0.46 ±0.02Ba | 8.44 ±1.31Ba |
| | 5-20 cm | 1.25 ±0.05Aa | 8.29 ±0.04Aa | 0.43 ±0.07Ba | 0.45 ±0.01Ba | 6.22 ±0.88Bb |

Note: Different capital letters mean significant difference at $p<0.05$ for the same soil
layer and different vegetation type.
Different small letters mean significant difference at $p<0.05$ for the same vegetation
type and different soil layers (n=72).

Table 3
The correlation coefficients between soil labile organic carbon content and enzyme
activities in 0-5 cm

| Correlation coefficients | SOC | MBC | EOC | DOC | Urease | Sucrase | Alkaline phosphatase |
|---|---|---|---|---|---|---|---|
| SOC | 1 | .928** | .843** | .579* | .823** | .907** | 0.512 |
| MBC | | 1 | .799** | 0.537 | .787** | .877** | 0.57 |
| EOC | | | 1 | .702* | .775** | .818** | .588* |
| DOC | | | | 1 | .886** | 0.568 | .727** |
| Urease | | | | | 1 | .738** | .717** |
| Sucrase | | | | | | 1 | 0.478 |
| Alkaline phosphatase | | | | | | | 1 |

Note: **. Significant relation at 0.01 levels = * .Significant relation at 0.05 levels




Table 4
The correlation coefficients between soil labile organic carbon content and enzyme
activities in 5-20 cm

| Correlation coefficients | SOC | MBC | EOC | DOC | Urease | Sucrase | Alkaline phosphatase |
|---|---|---|---|---|---|---|---|
| SOC | 1 | .902** | .855** | 0.18 | .762** | .759** | 0.57 |
| MBC | | 1 | .762** | 0.106 | .633* | .840** | .656* |
| EOC | | | 1 | 0.391 | .621* | .593* | .600* |
| DOC | | | | 1 | 0.414 | 0.222 | 0.229 |
| Urease | | | | | 1 | 0.503 | .680* |
| Sucrase | | | | | | 1 | 0.397 |
| Alkaline phosphatase | | | | | | | 1 |

Note: **. Significant relation at 0.01 levels = * .Significant relation at 0.05 levels
Table5
The canonical correlation coefficients (CCA) between soil carbon fractions and enzyme
activities

| Canonical correlation coefficient significance test | | | | | Proportion that can be explained (%) | | | |
|---|---|---|---|---|---|---|---|---|
| | | | | | X-CV | | Y-CV | |
| No. | Correlation | Chi-SQ | DF | Sig. | Within-cluster | Between-cluster | Within-cluster | Between-cluster |
| 1 | 0.964 | 59.543 | 12 | 0.000 | 0.743 | 0.690 | 0.728 | 0.677 |
| 2 | 0.537 | 9.273 | 6 | 0.159 | 0.052 | 0.015 | 0.076 | 0.022 |
| 3 | 0.370 | 2.800 | 2 | 0.247 | 0.122 | 0.017 | 0.196 | 0.027 |

$U_1=-0.558X_1-0.309X_2-0.108X_3-0.365X_4$
$V_1=-0.500Y_1-0.549Y_2-0.046Y_3$

Note: The CCA was performed using the soil carbon fractions and enzyme activities,
three pairs of canonical variates (CVs) were extracted. The $U_1$ and $V_1$ refer to the first
group equation between soil carbon fractions canonical variate (X-CV) and the
enzyme activities canonical variate (Y-CV) which has the highest significant
coefficients of <0.05. The rest two equations of $U_2$, $V_2$-$U_3$, $V_3$ did not show since its
canonical correlation coefficients were higher than 0.05.The indices of the X-CV were:
SOC (X1), MBC (X2), EOC (X3), and DOC (X4). The indices of the Y-CV were:
urease activity (Y1), sucrase activity (Y2), alkaline phosphatase activity (Y3).


Figure

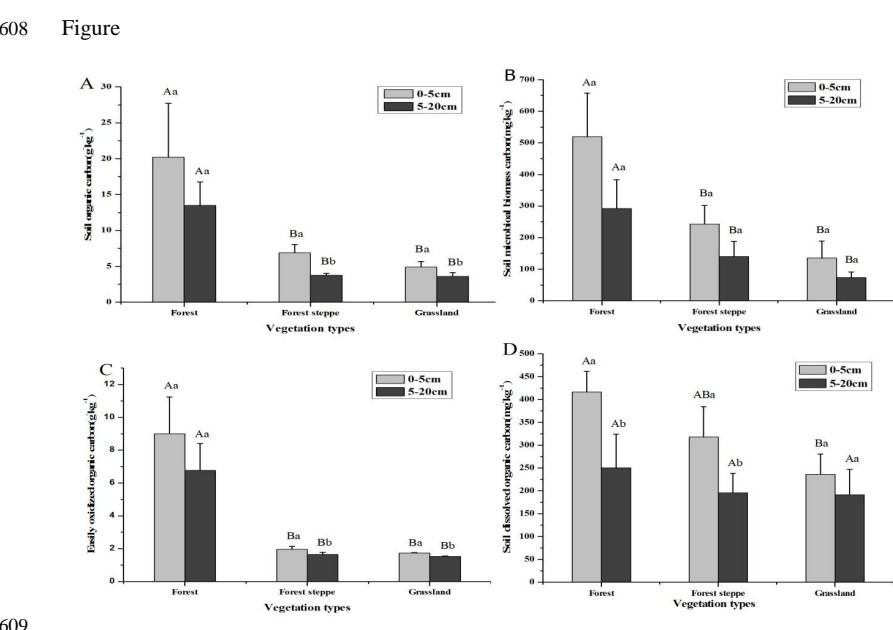


Fig.1. SOC (A), MBC (B), EOC (C) and DOC (D) under different vegetation types
Note: Different capital letters mean significant differences at p<0.05 for the same soil
layer and different vegetation type.
Different small letters mean significant differences at p<0.05 for the same vegetation
type and different soil layers (n=72).



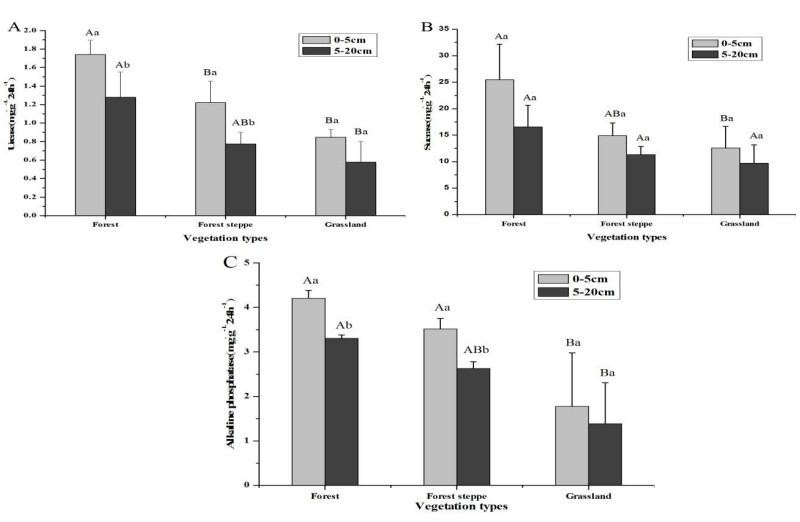


Fig.2. Soil urease (A), sucrase (B) and alkaline phosphatase (C) activity under
different vegetation types
Note: Different capital letters mean significant differences at p<0.05 for the same soil
layer and different vegetation type.
Different small letters mean significant differences at p<0.05 for the same vegetation
type and different soil layers (n=72).



