# Peer review of "Soil carbon fractions and enzyme activities under different vegetation types on the Loess Plateau of China"

_Solid Earth, 2016_

## Referee Comment (RC1) · Anonymous Referee #1 · 29 Nov 2016

General comments: The manuscript entitled "oil carbon fractions and enzyme activities under different vegetation types on the Loess Plateau of China" shows data of carbon fractions and enzyme activities under different vegetations. The manuscript lacks the novelty. In addition, it is difficult to be understood for the English is poor and the language is with lots of grammar errors and unclear expressions. I suggest the author ask for the English expert to polish the language.

Specific comments:

Introduction: 1. The authors did not address relevant scientific questions and the objects of this study. 2. Page 3, line 45, "soils health" should be changed to "soil health" 3. Page 5, line 94, "various type vegetations" is corrected with "various vegetation types".

[Figure]

Materials and Methods: 1. Page 6, lines 106-109, "For each vegetation type, four representative plant communities were chosen (Table 1), and, as replicates, three sampling areas were defined in the field for each representative plant community. In each representative plant community, three sampling plots were delineated. " What it means? 2. Page 6, line 121, "different type soil" is corrected with "different soil types".

Results 1. Page 10, lines 197-198, "The total N concentration of forest was significantly higher than the total N of both forest steppe and grassland" is corrected with "The total N concentration in forest was significantly higher than those in both forest steppe and grassland". 2. Page 10, lines 199-200, "The total P concentration of grassland vegetation was significantly lower than for both forest and forest steppe." This sentence is replaced with "The total P concentration was significantly lower in grassland than in both forest and forest steppe." 3. Page 10, line 223, "was" is corrected with "were".

Disscusion With respect to the effect of vegetation on soil carbon fractions and enzyme activities, there were some literatures. Why have the authors yet studied the effect?

---

## Short Comment (SC1) · 30 Nov 2016

The subject matter of the manuscript as well as the hypothesis has little novelty. It is well known that composition of SOC is related to enzymatic activity. It is obvious for the unaltered forest soil to be rich in C and enzyme activity. Moreover, paer entitled "Changes in soil nutrient and enzyme activities under different vegetations in the Loess Plateau area, Northwest China" Wang et al (2012) published in Catena 92 (2012) 186–195 also describes similar work in the same region. Measuring SOC in top soil (upto 20 cm) for trees is not a sensible option. The author should analyse it for much deeper regions.

---

## Referee Comment (RC2) · Anonymous Referee #2 · 7 Jan 2017

Soil carbon fraction and soil enzyme can indicate the impact of different vegetation on soil quality. This manuscript deals with a very interesting topic, and the study results is important and relevant to the subject coverage of this journal. The content of soil organic carbon and its fractions as well as the soil enzyme activity were measured under three vegetation types. So the paper provide some data, and this is useful for understanding the effect of three vegetation types. But the innovation of methodology and ideas are insufficient. The level of English throughout your manuscript does not meet the desired standard. Please check the manuscript and refine the language carefully.

Major comment: The topical subject is not clear. I have well understood the purpose of

this study was to compare the difference of soil organic carbon and soil enzyme activity among the three vegetation types. However, in the introduction section, it states "there is a lack of the information on the relationship between the soil carbon and enzyme activity". This is blurred as a study topic.

Author just stated "Vegetation type was an important factor influencing the variation of soil enzyme activities and carbon fractions on the Loess Plateau". Whereas which vegetation type is more beneficial to improve soil fraction or soil enzyme activity was ambiguous. So the conclusions are weak. The scientific design on different vegetation is reasonable. Whereas the second and third hypotheses in introduction are not specific. The sampling soil method in grassland is not reasonable. Author stated that the plot size for grassland is only $1 \times 1$ m, and 9 sub-samples were collected. How variable the results could be? I think the plot size is too small. In order to cover statistical tests, the plot size should be increased for sampling soil. The results and discussion are not well structured and documented. In the discussion section, more sentences are descriptive, and do not clearly support the objective of the study.

Minor comment: The title is not very clear, the word "impact" or "variation" or the other should be added. The abstract is well organized, whereas the conclusion miss points. The results are not informative. Despite the magnitude of the experimental work, the statistical analysis are not enough, and conclusions are not sufficiently substantiated. The author can try a multivariate analysis.

Some study methods should be clarified: How to remove the living grass in grassland when sampling soil. The more basic information on the three study sites (Fuxian, Ansai, Lian Daowan), such as topography, soil types, the management history on the different vegetation types need be reported and discussion. 4 representative plant community were selected under one vegetation type. How much is the variation of soil organic carbon and soil enzyme activity under these representative plant community. Fresh soil are recommended in some assay. Author stated that the soil sample was kept at -20°C. Whereas air-dried soil was adopted for measuring urease activity and

[Figure]

Soil DOC. Why. CCA is the common abbreviation of canonical correspondence analysis, and it is not proper to be used in line 179 and the following parts. Authors should clarify the abbreviations they used.

Some section should be reduced: In introduction section, the impact of vegetation restoration on soil property should be reduced, and enhance more substance on the effect of different vegetation types. Some unmodified assay process for soil organic carbon or soil enzyme measurement could be removed or reduced. In 3.4 section, I suggest that the both section of 3.4.0 and 3.4.2 should be combined. Author selected urease, sucrose and alkaline phosphatase. If more enzyme activities were measured, the manuscript quality will be enhanced.

---

## Short Comment (SC2) · 7 Jan 2017

Dear Editors and Reviewers: Thank you for your letter and for the reviewers' comments concerning our manuscript entitled "Soil carbon fractions and enzyme activities under different vegetation types on the Loess Plateau of China". Those comments are all valuable and very helpful for revising and improving our paper, as well as the important guiding significance to our researches. We have studied comments carefully and have made correction which we hope meet with approval. Revised portion are marked in blue in the paper. The main corrections in the paper and the responds to the Editor's and reviewer's comments are as flowing: Editor's comments:

–The subject matter of the manuscript as well as the hypothesis has little novelty. It is

well known that composition of SOC is related to enzymatic activity. It is obvious for the unaltered forest soil to be rich in C and enzyme activity. Moreover, paper entitled "Changes in soil nutrient and enzyme activities under different vegetations in the Loess Plateau area, Northwest China" Wang et al (2012) published in Catena 92 (2012) 186–195 also describes similar work in the same region. Measuring SOC in top soil (up to 20 cm) for trees is not a sensible option. The author should analyse it for much deeper regions. Response: We thank the reviewer's appreciation about the subject and hypotheses of our manuscript. We admit that, forest soils are known to be a strong C sink (in terms of C accumulation) as compared to those of the forest steppe and grassland soils, while the C fractions, especially the labile parts, they might exhibit various patterns along vegetation types. Compared to the work published by Wang et al (2012), the similar point is: One work was carried out on the Loess Plateau of China, the soil parameters such as nutrients as well as enzyme activities were analyzed, as a way to compare the difference between treatments. However, the obvious differences lies in the two studies are: 1) The objective of the study ïij∎ Wang et al., focused on evaluating the influence of land transformation (from former farmland to forestland) on the soil quality, after 3 decades. However, we emphasized on the relationship between soil C and enzyme activities along vegetation types. 2) In terms of soil C, Wang et al. only examined the content of soil organic matter, however, we investigated the four soil labile fractions (soil microbial biomass carbon, soil easily oxidized organic carbon and soil dissolved organic carbon), with a consideration of the essential role played by soil labile C in terms of soil nutrients turnover and enzyme activity regulation. We tried our best to improve the manuscript and made some changes in the manuscript. These changes will not influence the content and framework of the paper. We appreciate for Editors/Reviewers' warm work earnestly, and hope that the correction will meet with approval. Once again, thank you very much for your comments and suggestions.

Thank you and best regards. Yours sincerely,

Shaoshan An (Prof. Dr.) College of Natural Resources and Environment, Northwest

A&F University, 712100, P.R. China State key laboratory of soil erosion and dryland farming on the Loess Plateau, Institute of Soil and Water Conservation, Northwest A&F University, 712100, P.R. China

Please also note the supplement to this comment:
http://www.solid-earth-discuss.net/se-2016-137/se-2016-137-SC2-supplement.pdf

---

## Short Comment (SC3) · 7 Jan 2017

Dear Editors and Reviewers: Thank you for your letter and for the reviewers' comments concerning our manuscript entitled "Soil carbon fractions and enzyme activities under different vegetation types on the Loess Plateau of China". Those comments are all valuable and very helpful for revising and improving our paper, as well as the important guiding significance to our researches. We have studied comments carefully and have made correction which we hope meet with approval. Revised portion are marked in blue in the paper. The main corrections in the paper and the responds to the Editor's and reviewer's comments are as flowing: Editor's comments: –The manuscript lacks the novelty. The authors did not address relevant scientific questions and the

objects of this study. Response: Based on the four carbon fractions and three enzyme activities under various type vegetations considered in our experiments. In our opinion, this study, to some degreeïijŇenriches and broadens the vegetation ecology and restoration ecology of this area and provides important theoretical basis for ecosystem restoration and reconstruction in Loess Plateau of China. We agree with reviewer's suggestions that the objectives and hypotheses were not clearly stated in the former paper. The initial idea was to bridge the connection between soil carbon fractions and enzyme activities for soils with different vegetation types. Therefore, we emphasized the purposes of the experiment, and the new section has been added just before the Hypotheses in the INTRODUCTION part: "The research questions proposed in our study were i) whether the content of soil labile organic C fractions and three enzymes were higher in the soils of forest than in the forest steppe and grassland? ii) does three labile organic C fractions exert various effects to enzyme activities in soils of difference vegetation types? iii) does the three vegetation types exhibited a similar vertical change along soil profile? ïij∎ that is, the contents of the investigated soil parameters were decreasing from top to deeper soil layers." Three hypotheses formulated based on our scientific questions were presented at the end of the INTROCUTION part, now.

–Page 3, line 45, "soils health" should be changed to "soil health". Response: We changed "soils health" to "soil health" –Page 5, line 94, "various type vegetations" is corrected with "various vegetation types". Response: We changed "various type vegetations" to "various vegetation types" –Page 6, lines 106-109, "For each vegetation type, four representative plant communities were chosen (Table 1), and, as replicates, three sampling areas were defined in the field for each representative plant community. In each representative plant community, three sampling plots were delineated. " What it means? Response: -4 plant communities. - In each community, 3 sampling areas. - In each sampling area, three sampling plots. - In each plot. A composite sample (made of 9 sub-samples) for 0-5 cm and a composite sample for 5-20 cm –Page 6, line 121, "different type soil" is corrected with "different soil types" Response: We changed "different type soil" to "different soil types" –Page 10, lines 197-198, "The total N concentration of

forest was significantly higher than the total N of both forest steppe and grassland" is corrected with "The total N concentration in forest was significantly higher than those in both forest steppe and grassland" Response: We changed "The total N concentration of forest was significantly higher than the total N of both forest steppe and grassland" to "The total N concentration in forest was significantly higher than those in both forest steppe and grassland" –Page 10, lines 199-200, "The total P concentration of grassland vegetation was significantly lower than for both forest and forest steppe." This sentence is replaced with "The total P concentration was significantly lower in grassland than in both forest and forest steppe." Response: As suggested by the review, the former sentence "The total P concentration of grassland vegetation was significantly lower than for both forest and forest steppe." has been replaced by "The total P concentration was significantly lower in grassland than in both forest and forest steppe." –Page 10, line 223, "was" is corrected with "were". Response: We changed "was" to "were" –Disscusion With respect to the effect of vegetation on soil carbon fractions and enzyme activities, there were some literatures. Why have the authors yet studied the effect? Response: Although soil carbon fractions and enzyme activities are well-known, few studies have examined correlations among soil organic carbon, soil microbial biomass carbon, soil easily oxidized organic carbon, soil dissolved organic carbon and soil enzyme activities of different vegetation types in Loess Plateau of China. As for 4.2, the main focus has been shifted from the "4.2 Soil enzyme activities under different vegetation types" to "4.2 The vertical change of the soil enzyme activities", and we discussed the enzyme variations along soil depth gradients, and tried to explained the reason in terms of litter inputs, soil microorganisms, above-ground vegetation effects. To sum up, although soil carbon and enzyme activities are studied in previous works, there is a less focus on the relationship between soil enzyme activities and labile C fractions, namely soil microbial biomass carbon, soil easily oxidized organic carbon, soil dissolved organic carbon. Our results demonstrate differential effects of vegetation types on C fractions and enzyme activities further exhibiting dissimilar features among these observed effects on the Chinese Loess Plateau. We tried our best to improve the manuscript and made

some changes in the manuscript. These changes will not influence the content and framework of the paper. We appreciate for Editors/Reviewers' warm work earnestly, and hope that the correction will meet with approval. Once again, thank you very much for your comments and suggestions.

Thank you and best regards. Yours sincerely,

Shaoshan An (Prof. Dr.) College of Natural Resources and Environment, Northwest A&F University, 712100, P.R. China State key laboratory of soil erosion and dryland farming on the Loess Plateau, Institute of Soil and Water Conservation, Northwest A&F University, 712100, P.R. China

Please also note the supplement to this comment:
http://www.solid-earth-discuss.net/se-2016-137/se-2016-137-SC3-supplement.pdf

---

## Author Comment (AC1) · 13 Jan 2017

Dear Editors and Reviewers: Thank you for your letter and for the reviewers' comments concerning our manuscript entitled "Soil carbon fractions and enzyme activities under different vegetation types on the Loess Plateau of China". Those comments are all valuable and very helpful for revising and improving our paper, as well as the important guiding significance to our researches. We have studied comments carefully and have made correction which we hope meet with approval. Revised portion are marked in blue in the paper. The main corrections in the paper and the responds to the Editor's and reviewer's comments are as flowing:

Anonymous Referee #2 – The topical subject is not clear. I have well understood the

purpose of this study was to compare the difference of soil organic carbon and soil enzyme activity among the three vegetation types. However, in the introduction section, it states "there is a lack of the information on the relationship between the soil carbon and enzyme activity". This is blurred as a study topic. Response: Thank you very much for your suggestion. "there is a lack of the information on the relationship between the soil carbon and enzyme activity" has been replaced by "there is a lack of information on the relationship among SOC, MBC, EOC, DOC and enzyme activities for soils with forest, forest steppe and grassland vegetation types". – Author just stated "Vegetation type was an important factor influencing the variation of soil enzyme activities and carbon fractions on the Loess Plateau". Whereas which vegetation type is more beneficial to improve soil fraction or soil enzyme activity was ambiguous. Response: We added "Forest vegetation type was more beneficial to improve soil fraction and soil enzyme activity." in line 381-382. – The scientific design on different vegetation is reasonable. Whereas the second and third hypotheses in introduction are not specific. Response: We changed hypotheses to "i) whether the content of soil labile organic C fractions and three enzymes were higher in the soils of forest than in the forest steppe and grassland? ii) does three labile organic C fractions exert various effects to enzyme activities in soils of difference vegetation types? iii) does the three vegetation types exhibited a similar vertical change along soil profile?" – The sampling soil method in grassland is not reasonable. Author stated that the plot size for grassland is only 1×1 m, and 9 sub-samples were collected. How variable the results could be? I think the plot size is too small. Response: Thank you very much for your valuable suggestion. Within each plot, based on an S-shaped sampling pattern, the incompletely-degraded litter was removed and 9 sub-samples were simultaneously and randomly collected by soil auger (20 cm×5 cm), then mixed them in the same bag which as a representative soil sample, separately at 0-5 cm and 5-20 cm depth. And the plot size for grassland was 1×1 m which also appeared in other paper (Cheng et al., 2015). – The title is not very clear, the word "impact" or "variation" or the other should be added. Response: We changed title to "Variations of soil carbon fractions and enzyme activities under different vegetation types on the Loess Plateau of China". – The abstract is well organized, whereas the conclusion miss points. Response: We added "Forest vegetation type was more beneficial to improve soil fraction and soil enzyme activity." in abstract. – How to remove the living grass in grassland when sampling soil. Response: The living grass was cut off by scissor when sampling soil, then each soil was sieved (2 mm) to remove large roots, stones and the macrofauna. –The more basic information on the three study sites (Fuxian, Ansai, Lian Daowan), such as topography, soil types, the management history on the different vegetation types need be reported and discussion. 4 representative plant communities were selected under one vegetation type. Response: Soils are described as Calcaric Cambisols according to the FAO classification system in our study sites (Jiao et al., 2013). And the other information should be found in Table 1. – How much is the variation of soil organic carbon and soil enzyme activity under these representative plant communities. Response: At 0-5 cm soil layer, SOC, MBC, EOC and DOC contents of forest vegetation were 2.93, 2.14, 4.60 and 1.31 times compared with forest steppe. And SOC, MBC, EOC and DOC contents of forest vegetation were 4.12, 3.85, 5.18 and 1.76 times compared with grassland vegetation. Analogously at 5-20 cm soil layer, SOC, MBC, EOC and DOC contents of forest vegetation were 3.58, 2.09, 4.10 and 1.28 times compared with forest steppe. And SOC, MBC, EOC and DOC contents of forest vegetation were 3.73, 4, 4.42 and 1.31 times compared with grassland vegetation. At 0-5 cm soil layer, urease activity of forest vegetation was 1.43 and 2.05 times compared with forest steppe and grassland vegetation, analogously at 5-20 cm soil layer, sucrase activity of forest vegetation was 1.65 and 2.21 times by comparing with forest steppe and grassland vegetation. At 0-5 cm soil layer, sucrase activity of forest vegetation was 1.71 and 2.03 times compared with forest steppe and grassland vegetation, analogously at 5-20 cm soil layer, sucrase activity of forest vegetation was 1.46 and 1.70 times by comparing with forest steppe and grassland vegetation. At 0-5 cm soil layer, alkaline phosphatase activity of forest vegetation was 1.20 and 2.38 times compared with forest steppe and grassland vegetation, analogously at 5-20 cm soil layer, alkaline phosphatase activity of forest vegetation was

1.25 and 2.39 times by comparing with forest steppe and grassland vegetation. – Fresh soils are recommended in some assay. Author stated that the soil sample was kept at -20âŮęC. Whereas air-dried soil was adopted for measuring urease activity and Soil DOC. Response: Thank you very much for your suggestion. The representative soil sample was split into two parts, one was stored intact at -20°C in order to determine MBC, and the other was air-dried for measuring soils' enzyme, physics and chemical properties. – CCA is the common abbreviation of canonical correspondence analysis, and it is not proper to be used in line 179 and the following parts. Response: We changed "A canonical correlation coefficients analysis (CCA)" to "Canonical correlation analysis (CCA)" (Huang et al., 2015) – In 3.4 section, I suggest that the both section of 3.4.1 and 3.4.2 should be combined. Response: Thank you very much for your valuable suggestion. Both section of 3.4.1 and 3.4.2 had been combined. – Some section should be reduced: In introduction section, the impact of vegetation restoration on soil property should be reduced, and enhance more substance on the effect of different vegetation types. Response: Thank you very much for your suggestion. "Recently, some studies have concentrated on the vegetation restoration, for instance, Jiao et al., (2011) found that revegetation had positive effects on the soil physical properties. In the protected vegetation areas, relative humidity of air increased and wind velocity is greatly reduced. Additionally, bulk density of the surface layer (0-20 cm) significantly decreases while soil porosity, water-holding capacity, aggregate stability, and saturated hydraulic conductivity significantly increase. SOC stocks are increased by 19% in the surface soil layer at 0-20 cm soil depth from 1998 to 2006, because of the vegetation restoration in the Loess Plateau (Wang et al., 2011)." this part was removed. "Cheng et al. (2015) investigated shrubland CK16 (16-year-old Caragana korshinskii Kom.), shrubland CK26 (26-year-old C. korshinskii Kom.), shrubland AS (Armeniaca sibirica Lam.), natural grassland and artificial pasture vegetations, she found that conversion to C. korshinskii shrublands and protection of natural grassland should be promoted to improve the contribution of vegetation to SOC sequestration. Fu et al. (2010) selected transitional belt. Korshinsk Peashrub (KOP), purple alfalfa (ALF), natural fallow (NAF)

and millet (MIL) vegetation types, and found that ALF and NAF compared with MIL, did not show much potential to increase SOC in study. Alpine swamp meadow (ASM) was compared with alpine meadow (AM), alpine steppe (AS) and alpine desert (AD) vegetations, it was the best system conserving soil nutrient (especially labile fractions) and microbial activity in permafrost regions of the China's Qinghai-Tibet Plateau (Shang et al., 2016). The Wang et al. (2012) showed that after 30 years of restoration, nutrients content in the soil of mixed forest of black locust and amorpha increased significantly. However, nutrients content in the soil of mixed forest of Chinese pine and amorpha decreased. As to soil enzyme activities increased while polyphenol oxidase activity decreased compared to non-restoration and climax community soils." this part was added in introduction section – In the discussion section, more sentences are descriptive, and do not clearly support the objective of the study. Response: Thank you very much for your suggestion. "In the protected vegetation areas, relative humidity of air increased and wind velocity is greatly reduced. Additionally, bulk density of the surface layer (0-20 cm) significantly decreases while soil porosity, water-holding capacity, aggregate stability, and saturated hydraulic conductivity significantly increase (Jiao et al., 2011)." and "SOC stocks are increased by 19% in the surface soil layer at 0-20 cm soil depth from 1998 to 2006, because of the vegetation restoration in the Loess Plateau (Wang et al., 2011)." were added in the discussion section.

ReferencesïijŽ Cheng, M., Xue, Z. J., Xiang, Y., Darboux, F. and An, S.S.: Soil organic carbon sequestration in relation to revegetation on the Loess Plateau, China, Plant and Soil, 397,31-42,2015. Huang, Y. M., Liu, D., An, S. S.: Effects of slope aspect on soil nitrogen and microbial properties in the Chinese Loess region, Catena, 125, 135-145, 2015. Jiao, F., Wen, Z. M., and An, S. S.: Changes in soil properties across a chronosequence of vegetation restoration on the Loess Plateau of China, Catena, 86, 110-116, 2011. Jiao, F., Wen, Z. M., and An, S. S and Yuan, Z.: Successional changes in soil stoichiometry after land abandonment in Loess Plateau, China, Ecological Engineering, 58, 249-254, 2013. Shang, W., Wu, X. D., Zhao,L., Yue, G. Y., Zhao, Y. H., Qiao, Y. P. and Li, Y. Q.: Seasonal variations in labile soil organic matter

fractions in permafrost soils with different vegetation types in the central Qinghai-Tibet Plateau, Catena , 137, 670-678, 2016. Fu, X. L., Shao, M. A., Wei, X. R., and Horton, R.: Soil organic carbon and total nitrogen as affected by vegetation types in Northern Loess Plateau of China, Geoderma, 155 , 31-35, 2010. Wang, B., Xue, S., Liu, G. B., Zhang, G. H., Li, G. and Ren, Z. P.: Changes in soil nutrient and enzyme activities under different vegetations in the Loess Plateau area, Northwest China, Catena, 92, 86-95, 2012. Wang, Y., Fu, B., Lü, Y., and Chen, L.: Effects of vegetation restoration on soil organic carbon sequestration at multiple scales in semi-arid Loess Plateau, China, Catena 85, 58-66, 2011.

We tried our best to improve the manuscript and made some changes in the manuscript. These changes will not influence the content and framework of the paper. We appreciate for Editors/Reviewers' warm work earnestly, and hope that the correction will meet with approval. Once again, thank you very much for your comments and suggestions.

Thank you and best regards. Yours sincerely, Shaoshan An (Prof. Dr.) College of Natural Resources and Environment, Northwest A&F University, 712100, P.R. China State key laboratory of soil erosion and dryland farming on the Loess Plateau, Institute of Soil and Water Conservation, Northwest A&F University, 712100, P.R. China

Please also note the supplement to this comment:
http://www.solid-earth-discuss.net/se-2016-137/se-2016-137-AC1-supplement.pdf